

# PERSONALIZED VISUAL INSTRUCTION TUNING

**Renjie Pi**[1]*, **Jianshu Zhang**[1]*, **Tianyang Han**[1], **Jipeng Zhang**[1], **Rui Pan**[2], **Tong Zhang**[2]

[1]The Hong Kong University of Science and Technology   [2]University of Illinois Urbana-Champaign
{rpi,jzhanggr}@ust.hk, jianshu.zhang@whu.edu.cn, 23104841g@connect.polyu.hk,
ruip4@illinois.edu, tongzhang@tongzhang-ml.org

## ABSTRACT

Recent advancements in multimodal large language models (MLLMs) have demonstrated significant progress; however, these models exhibit a notable limitation, which we refer to as "face blindness". Specifically, they can engage in general conversations but fail to conduct personalized dialogues targeting at specific individuals. This deficiency hinders the application of MLLMs in personalized settings, such as tailored visual assistants on mobile devices, or domestic robots that need to recognize members of the family. In this paper, we introduce **Personalized Visual Instruction Tuning (PVIT)**, a novel data curation and training framework designed to enable MLLMs to identify target individuals within an image and engage in personalized and coherent dialogues. Our approach involves the development of a sophisticated pipeline that autonomously generates training data containing personalized conversations. This pipeline leverages the capabilities of various visual experts, image generation models, and (multi-modal) large language models. To evaluate the personalized potential of MLLMs, we present a benchmark called P-Bench, which encompasses various question types with different levels of difficulty. The experiments demonstrate a substantial personalized performance enhancement after fine-tuning with our curated dataset.

## 1 INTRODUCTION

The advent of Large Language Models (LLMs) has significantly advanced AI, transforming natural language processing and understanding (Geng & Liu, 2023; OpenAI, 2023; Touvron et al., 2023; Scao et al., 2022; Chowdhery et al., 2022; Taori et al., 2023; Chiang et al., 2023). These models, trained on extensive text corpora, possess substantial world knowledge, excelling in various tasks. Progress in LLMs has led to rapid enhancements in Multimodal Large Language Models (MLLMs) (Liu et al., 2023a; Zhu et al., 2023; Dai et al., 2023; Li et al., 2023b; OpenAI, 2023; Bai et al., 2023). MLLMs leverage pretrained visual encoders (e.g., vision transformers) to process images, incorporating them as token embeddings alongside text token embeddings. These models expand LLM capabilities to engage in conversations based on image inputs, offering diverse applications like autonomous driving (Ding et al., 2023) and medical assistants (Li et al., 2023a).

Despite the success of MLLMs, their effectiveness is limited in general purpose conversations, and drastically fail at personalized conversations targeting at specific individuals. For example, given a photograph of a girl named Lisa, and an image with Lisa inside of it, the state-of-the-art MLLMs are not able to recognize her and provide corresponding information. This deficiency prohibits the use of the MLLMs in personalized use cases, such as your personal AI assistant deployed on the mobile phone, a domestic robot that needs to recognize and serve your family members, or a smart home system that requires individualized interactions to cater to the specific needs of different residents.

---

* Equal Contribution.
Code and data are available at the following links:
https://github.com/sterzhang/PVIT
https://huggingface.co/datasets/Sterzhang/PVIT-3M
The code and data are released under MIT and apache2.0 licenses, respectively.

To empower MLLMs with personalization capability (denoted as P-MLLM), previous endeavors propose to augment the MLLM with external heads and vocabularies, which are trained to identify specific individuals within a scene using a few personalized training data (Alaluf et al., 2024; Nguyen et al., 2024). Although these approaches demonstrate good performances, they suffer from the following weaknesses: 1) they require additional training for each newly introduced individual, which is inflexible and unpractical for real-life scenarios, since the person of interest may frequently change; 2) it can not be guaranteed that we can always collect the training data for the person of interest. Therefore, it is undoubtedly more promising if the personalization capability of the MLLM is able to generalize, rather than being limited on a few pre-defined individuals.

Owing to the auto-regressive training paradigm adopted for current state-of-the-art LLMs and MLLMs, they possess the ability to generate responses depending on a given prefix. This ability is also referred to as in-context learning capability (Wei et al., 2023), which enables the model to produce different outputs during inference by adjusting the prefix without further training. Therefore, one intuitive and practical approach is to provide the information of individuals to the MLLM in its prefix. In this way, the MLLM is expected to provide answers for different input individuals during inference. However, our experimental results indicate that this capability is challenging, and current MLLMs struggle to effectively comprehend such personalized inputs. This may be due to the fact that these MLLMs are fine-tuned with limited multimodal data and a lack of personalized data, which together hinder their ability to develop in-context understanding for multimodal inputs.

To address this challenge, we propose **Personalized Visual Instruction Tuning (PVIT)**, a novel training paradigm that enables MLLMs to perceive personalized inputs as in-context prefixes. Specifically, each individual is represented as a <personal image, personal introduction> pair, which is provided to the MLLM as a *multi-modal prefix*. We further introduce *personalized wrapper tokens* to group the visual and textual information of each individual, thereby eliminating ambiguity when multiple individuals are involved. During training, the MLLM is optimized to answer questions related to target individuals within the prefixes. Once trained, the MLLM is able to fully utilize its in-context learning capability, and generalizes to arbitrary individuals without requiring additional fine-tuning or modifications to the model architecture.

In our proposed PVIT paradigm, the most critical barrier is the absence of large-scale, high-quality training data. To address this challenge, we design an automatic framework to synthesize personalized training data, operating in three phases. First, visual expert models extract individual visual concepts from scene images (*Visual Concept Curation*). Next, we utilize the MLLM to convert these visual concepts into both individual-level and scene-level textual descriptions, which are fused to create a coherent representation (*Dual-Level Textual Information Extraction and Fusion*). Finally, LLMs generate diverse personalized QA pairs using reasoning and instruction-following capabilities (*PVIT Dataset Generation*). To evaluate the personalization capability of MLLMs, we further created a benchmark termed **P-Bench**, which assesses personalization capability from multiple perspectives. The results indicate that the ability of current SOTA MLLMs to perceive personalized concepts is limited, which can be significantly boosted after training with our proposed PVIT.

To summarize, we make the following contributions in this paper:

- Inspired by in-context learning capabilities of LLMs, we propose **Personalized Visual Instruction Tuning (PVIT)**, a new training paradigm that equips MLLMs to conduct personalized conversation for any arbitrary individuals with no addition training at inference.

- We meticulously design an automatic data annotation framework to curate high-quality personalized training data, and synthesize a large scale dataset to enhance the MLLM's capability to conduct personalized conversations.

- We curate **P-Bench**, a novel benchmark for evaluating the personalization capability of MLLMs. We demonstrate that the MLLM trained with our curated dataset demonstrates significantly improved performances.

## 2 RELATED WORK

**Multi-Modal Large Language Model.** Recent advancements in large language models (LLMs) have significantly improved language comprehension and generation, achieving near-human profi-

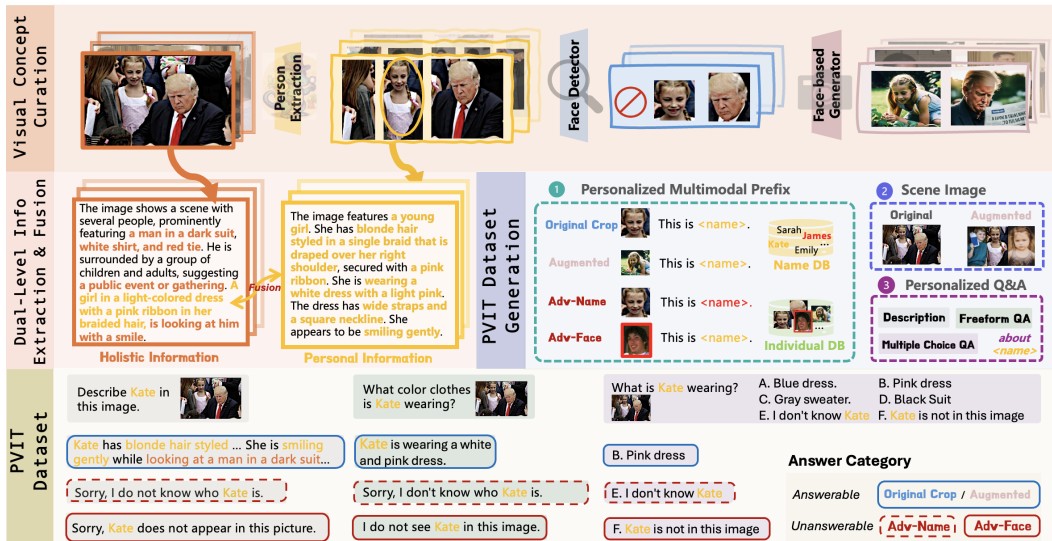

Figure 1: The **Personalized Visual Instruction Tuning (PVIT)** framework consists of three phases. In the *visual concept curation* phase, we extract individuals and their faces from images, then augment them with different poses and angles. During the *dual-level textual information extraction and fusion* phase, MLLMs first generate both holistic information and personal information, then integrate them to get more detailed and contextually accurate information. In the *PVIT dataset generation* phase, LLMs create QA pair templates based on the extracted textual information, which are filled with selected names to construct training data.

ciency across various tasks (Brown et al., 2020; Scao et al., 2022; Chowdhery et al., 2022; Smith et al., 2022; Hoffmann et al., 2022; Ouyang et al., 2022; Touvron et al., 2023; Bai et al., 2022). This success has spurred interest in vision-language interaction, leading to multi-modal large language models (MLLMs) (Liu et al., 2023a; Li et al., 2023b; Dai et al., 2023; Zhu et al., 2023; Dai et al., 2023; OpenAI, 2023; Bai et al., 2023; Su et al., 2023; Gao et al., 2023; Pi et al., 2023), which excel in dialogue based on visual inputs. However, they struggle with personalized conversations, such as dialogues about specific individuals like a girl named Kitty, limiting their use in personalized AI assistants, tailored recommendations, or customized therapy. Addressing this limitation is crucial for enabling personalized AI interactions.

**Model Personalization** In model personalization, the main objective is to tailor a model to learn new user-specific concepts. Considerable research has been conducted in the area of text-to-image (t2i) generation personalization (Gal et al., 2023; Ruiz et al., 2022; Alaluf et al., 2023; Arar et al., 2023; Voynov et al., 2023; Ye et al., 2023; Wang et al., 2024). The predominant method for t2i personalization involves fine-tuned the word embeddings using a few examples to capture the nuances of the target concept. Alternatively, some research has concentrated on personalizing image captioning models (Chunseong Park et al., 2017; Zeng et al., 2019; Wang et al., 2023; Park et al., 2018; Zeng et al., 2019; Shuster et al., 2019). These personalized captioning methods aim to produce captions in a particular writing style. In contrast, our goal is to enable the model to integrate a new user-specific concept into a personalized textual description response that aligns with a given image.

**Personalized MLLMs** Compared to text-to-image generation and traditional image captioning, the personalization of Multimodal Large Language Models (MLLMs) remains an under-explored area. Recent approaches propose introducing new tunable parameters for each new individual and conduct tuning using a few training samples. Specifically, MyVLM (Alaluf et al., 2024) adopts a Q-former-style architecture, incorporating learnable heads to extract specific concepts and append them to the visual features. On the other hand, Yo'LLaVA (Nguyen et al., 2024) utilizes an LLaVA-style architecture, proposing to directly incorporate new concepts as additional tokens in the LLM's vocabulary. Despite these approaches demonstrating promising performance in personalized conversation, they require additional training and parameters for each newly introduced individual. This requirement makes the paradigm less practical in real life since new individuals are likely to be incorporated from

time to time. In addition, it is often not possible to collect training samples associated with each individual. These limitations necessitate the model to generalize on new concepts on the fly.

## 3 PROBLEM FORMULATION

Unlike previous methods for personalized multimodal large language models (P-MLLMs), which require adding trainable parameters and fine-tuning for each new individual, our approach treats personalization as an in-context learning task. This eliminates the need for fine-tuning on each new individual. More specifically, we first provide the list of **multi-modal prefixes** to the MLLM, which consists of <personal image, personal introduction> pairs. Then, we present the **scene image** and a **personalized query** to conduct conversation targeting at specific individuals. Conditioned on the multi-modal prefixes and personalized query, the MLLM is expected to produce responses specific to the individual of interest accordingly.

We observe that this capability is nontrivial, which is not possessed by the majority of state-of-the-art MLLMs. As shown in Figure 2, the MLLM tends to produce general responses, even when the user instruction is targeting at a specific individual. This issue persists even in MLLMs trained on interleaved image and text data. We presume that this limitation is due to the lack of training, rather than deficiency in the visual encoder.

Therefore, we propose to empower the MLLM with the personalization capability via **Personalized Visual Instruction Tuning** (**PVIT**). Specifically, we first collect a training dataset $\mathcal{D} = \left\{ \left( r^i, q^i, I_s^i, \left\{ I_{p_k}^i, t_{p_k}^i \right\}_{k=1}^{K^i} \right) \right\}_{i=1}^N$, where $r^i$, $q^i$ and $I_s^i$ stand for the $i^{th}$ target response, user query, and the scene image provided to the MLLM, respectively. On the other hand, $\left\{ I_{p_k}^i, t_{p_k}^i \right\}_{k=1}^{K^i}$ represent the $K^i$ multimodal prefixes containing images of input individuals and their personalized introduction, such as names. The optimization problem can be formulated as follows:

$$\mathcal{L}(\mathcal{D}) = -\frac{1}{N} \sum_{i=1}^N \sum_{s=1}^{L^i} \log p \quad \left[ r^{i,s} | \mathcal{F}(r^{i,(<s)}, q^i, I_s^i, \left\{ I_{p_k}^i, t_{p_k}^i \right\}_{k=1}^{K^i}) \right], \tag{1}$$

During training, we minimize the auto-regressive loss to generate personalized responses based on the query, scene image, and multimodal prefixes. This approach fully harnesses the in-context learning capabilities of MLLMs, allowing them to adapt to new individuals without additional training.

**Personalized Wrapper Tokens** Naively interleaving personalized images and introduction in the multimodal prefixes may introduce ambiguity, as personalized introduction (e.g., a name) could be mistakenly associated with either the preceding or following person's image. This ambiguity complicates training and confuses the MLLM during inference. To resolve this issue, we introduce two special tokens into the MLLM's vocabulary: $\langle| \text{ person\_start } |\rangle$ and $\langle| \text{ person\_end } |\rangle$. These tokens serve as wrappers to clearly enclose each individual's information, structured as follows: $\langle| \text{ person\_start } |\rangle$ {photo} {text\_intro} $\langle| \text{ person\_end } |\rangle$. This design ensures that the information belonging to each individual is properly grouped, improving the model's learning process.

The main challenge of the in-context learning paradigm lies in the lack of personalized training data, as existing multimodal instruction-tuning datasets focus on general conversations. To address this, we propose an automated framework for annotating multimodal conversation data with personalized inputs, leveraging LLMs, MLLMs, image generation models, and vision experts. Details of this framework are discussed in the next section.

## 4 DATA CONSTRUCTION

To equip the MLLM with personalized conversation capabilities, we developed a data generation framework that synthesizes various types of personalized training data. The framework begins by utilizing a set of images, which serve as scene images for the training data. It first extracts visual concepts of the individuals from these images using vision expert models. Then, it converts the visual information contained in the images into textual descriptions using MLLMs. Finally, it utilized the LLM's reasoning and instruction following capability to create high-quality and diverse QA pairs. As

illustrated in Figure 1, the framework operates in three main phases: 1) Visual Concept Curation, 2) Dual-Level Textual Information Extraction and Fusion, and 3) PVIT Dataset Generation. Below, we describe the procedure for each phase in detail.

## 4.1 VISUAL CONCEPT CURATION

In this phase, based on the scene images, we devise strategies for collecting images of individuals. First, we apply **person identification** to accurately locate the individuals within the images. Next, we perform **person augmentation** to generate images of the same individuals in various contexts and different perspectives. These images will later serve as the foundation for creating both visual and textual information of the input individuals, as well as personalized QA pairs.

**Person Identification**   For each image, we apply an open-vocabulary object detector, such as GroundingDino (Liu et al., 2023b), to localize individuals by providing the image and the text prompt "a person". Next, for each detected person, we use a face detector (Geitgey, 2016) to identify and localize the corresponding face, as the face is the most distinctive feature for identification. The images of the individuals along their faces are stored for later stages. Individuals without a detected face are excluded during this process.

**Person Augmentation**   After the previous step, we derive a list of <person, face> pairs for each image. Each person in the scene image can be referenced by their corresponding face. However, in practice, the face used as reference to an individual often differs from the scene image. To introduce more variation in human faces and enhance the capability of MLLMs to recognize individuals, we adopt the identity preserving image generator PhotoMaker (Li et al., 2023c) to augment the person, which produces images of the same individual from different perspectives and contexts based on an input face. These augmented images can then be leveraged as reference to the individuals in the original scene images.

## 4.2 DUAL-LEVEL TEXTUAL INFORMATION EXTRACTION AND FUSION

To construct personalized conversations for specific individuals in the subsequent phase, it is essential to not only derive the characteristics of each person in the scene image, but also capture how they are interacting with the surrounding context, which will serve as the basis for creating conversations that accurately reflect each individual's role and behavior in the scene image. To achieve this, we employ a dual-level information extraction and fusion approach.

**Personal Information Extraction**   Since current MLLMs are unable to directly provide specific features of a designated person in the scene image containing more than one person, we focus on each individual by providing their cropped images, which are extracted from the previous phase, to the MLLM to create *personal information*. Since the cropped images contain only one person, the descriptions generated by the MLLM will only focus on the characteristics of this individual, which capture more fine-grained personalized details.

**Holistic Information Extraction**   As shown in Figure 1, although the personal features of the girl, such as "single braid", can be captured through the previous step, without the holistic context, it would be impossible to extract the information "the girl is looking at the man". Therefore, We utilize the existing descriptive capabilities of MLLMs to provide a *holistic information* of the scene image. Specifically, we emphasize describing the main characters in the image, such as the holistic information of describing "a man" and "a girl" in Figure 1. This approach aims to offer more "feature anchors" that facilitate the subsequent fusion of personal information and holistic information.

**Dual-Level Information Fusion**   After obtaining *holistic information* that provides a contextual knowledge and *personalized information* that captures individual characteristics, we attempt to link these two pieces of information. This is done by matching the personal information with the descriptions of characters in the holistic information, which results in a fused description that describes how a specific individual interacts with the context (demonstrated in Appendix 8). This dual-level fused information serves as the foundation for generating personalized conversations that are more detailed and contextually accurate.

## 4.3 PVIT DATASET GENERATION

With the visual concepts and the textual information associated with each individual extracted in the previous two phases , we can now construct the Personalized Visual Instruction Tuning (PVIT) dataset. The PVIT dataset primarily consists of three components: *Personalized Multimodal Prefix*, *Scene Image*, and *Personalized QA*.

### 4.3.1 PERSONALIZED MULTIMODAL PREFIX

In our in-context learning formulation of the personalization task, each input individual is represented by a multimodal prefix, which is the combination of a personal image and a personalized introduction. The personal image can be drived from the previous stage, which can either be the headshot cropped from the original image, or a generated photo from Photomaker (Li et al., 2023c). We design strategies to diversify the coverage of personalized introduction.

**Name Swapping**   We introduce <name> as a placeholder to be replaced with actual names during the construction of the training dataset. Specifically, we collect a list of names (around 600 names) using ChatGPT. Then, we randomly select a name and swap it with the placeholder. It is important to note that this process can be repeated multiple times, augmenting the training data with diverse names. This approach not only enhances the model's generalization ability and robustness in adapting to new individuals, but also helps prevent overfitting by avoiding the association of a specific person with a fixed name during training. The effectiveness of this technique is verified in Section 6.2.3.

**Personal Pronoun**   To better align with the way users refer to others in everyday conversations, we not only introduce individuals by their names but also handle situations involving personal pronouns (e.g., I, you, him). For example, if a question contains "my dad," the response should adapt by using "your dad." To handle such cases effectively, we introduce training data that includes examples of personal pronouns, ensuring the model can appropriately adjust its responses based on context.

**Adversarial Introduction**   To ensure the model truly learns to recognize individuals accurately, solely considering the cases where the questions are answerable is insufficient. The model must also learn to handle challenging or misleading scenarios. We found that even SOTA MLLMs often generate responses completely ignoring the input individuals provided in the prefix. For instance, even when the person of interest (e.g., a girl named Lisa) is not present in the image, model may still respond to questions about her and mistakenly identify other individuals as her. To address this issue, we introduce adversarial inputs designed to challenge the model's ability to correctly handle unanswerable questions. Specifically, we generate the following types of adversarial inputs:

- **Adversarial name mapping**: When choosing the person of interest to construct the query (e.g., Lisa), we make sure that the person is not provided at all in the personalized multimodal prefixes. For this type of query, the model should respond with "Sorry, I do not know who Lisa is", as the person was never introduced.
- **Adversarial image mapping**: When constructing the multimodal prefixes, we randomly select images of person excluding those that are in the scene image. The person of interest is one of the individual contained in the prefixes. In this case, the scene image does not contain this person of interest. The model should then respond with "Sorry, I cannot see Lisa in the image", demonstrating its ability to correctly identify missing individuals.

### 4.3.2 SCENE IMAGE

In designing scene images containing specific individuals, our goal is to enable the model to accurately recognize the target individuals provided in the prefix. We design two types of scene images for this purpose. The first type is the original, complete image, with the full context containing additional elements besides the person. This allows subsequent Q&A to be more comprehensive, focusing on both the specific features of individuals and their broader interactions with the environment. Additionally, to enhance the model's ability to recognize a specific person in an image containing multiple individuals, we also concatenate cropped images of individuals to create composite images, which elevates the MLLM's capability in more challenging scenarios.

### 4.3.3 PERSONALIZED QA

The dual-level information extracted from the previous stage transform visual information into texts, enabling us to utilize the LLM's advanced reasoning capabilities to create personalized conversations. Provided with such information, we meticulously design prompts and in-context examples to generate data for the following tasks using the LLM:

- **Personalized Description**: We create conversations where the user queries for the description of specific individuals, rather than the entire image.

- **Personalized Free-form QA**: We create free-form QA pairs that queries for information or characteristics related to specific individuals, such as appearance and behavior, which aims at enabling the MLLM to conduct personalized multi-round conversation.

- **Personalized Multi-choice QA** We create personalized QA pairs in multi-choice format, where the choices include the ground truth answer, as well as some confounding alternatives.

Compared with general purpose image captioning and VQA, the personalized conversation presents novel challenges for current MLLMs, since they not only need to recognize the individuals of interest in the scene image, but also properly incorporate them into the generated response.

In total, we create 3M training instances of personalized conversation, which we term **PVIT-3M**. The curated dataset encompasses diverse types of data and difficulty levels. We associate the detailed information and statistics of PVIT-3M in Section C of the Appendix..

## 5 EVALUATION USING P-BENCH

Although numerous benchmarks have been proposed to assess the effectiveness of MLLMs, none have been specifically designed to evaluate their personalization capabilities. To address this gap, we present a high-quality benchmark manually checked by human, **P-Bench**, aimed at thoroughly evaluating the personalization potential of MLLMs. We design both multiple choice (MC) questions and personalized image description queries for evaluation. In this section, we provide detailed overview of the problem types and evaluation metrics of P-Bench. Detailed statistics and curation process of the benchmark are presented in Section E.

### 5.1 MULTIPLE-CHOICE (MC) QUESTIONS

We design positive (answerable) and adversarial (unanswerable) questions to examine the MLLM's ability to correctly associate the target individual with the corresponding person in the scene image.

**Answerable Questions** include the following types: 1) *Crop*: the input individual is represented by its original face cropped from the image; 2) *Aug-In*: using Photomaker (Li et al., 2023c), we generate an augmented photo of individuals based on the original cropped face; 3) *Aug-Sc-2* and *Aug-Sc-3*: we concat two or three different cropped images of individuals into a single image to replace the original scene image, increasing the difficulty of accurately recognizing the individual.

**Unanswerable Questions**, on the other hand, include: 1) *Adv-name*: the question pertains to a person who is not included in the list of input individuals, meaning the MLLM lacks knowledge of this person; 2) *Adv-image*: the individual mentioned in the question does not appear in the scene image, meaning the MLLM cannot visually identify this person.

**Evaluation Metrics**   For MC questions, we adopt accuracy for the evaluation metric. To gain a deeper understanding of the MLLM's capabilities and limitations, we separately evaluate each of the three types in MC questions. In addition, to further study the MLLM's ability to differentiate different individuals, we also separately evaluate the accuracy of images containing different numbers of people.

### 5.2 DESCRIPTIVE QUESTIONS

For this type of questions, we query for the descriptions of specific individuals, rather than general descriptions. We also design both positive and adversarial description questions. Specifically, the

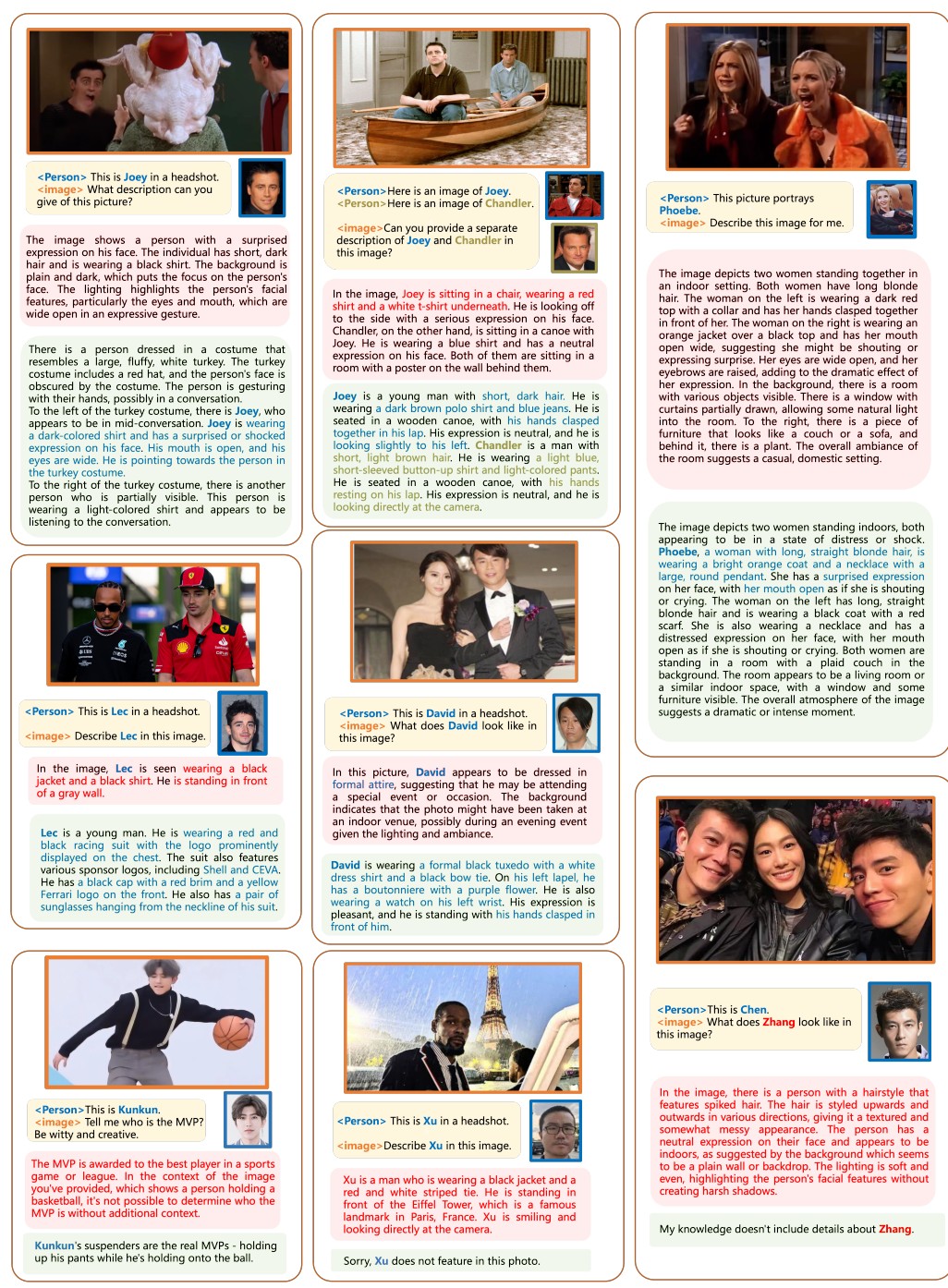

Figure 2: Qualitative examples of P-LLaVA results: Each example includes the user's query, input individual photos, and the scene image. The current MLLMs fail to recognize the person of interest and conduct personalized conversations, whereas our model, after training with PVIT, enables coherent and accurate personalized dialogues. Examples illustrate both answerable and unanswerable scenarios. For answerable cases, inputs involve single or multiple individuals, and our model incorporates names from the prefix for personalized responses. In unanswerable cases, current MLLMs provide incorrect answers, while the our model appropriately refuses and explains the reason.

Table 1: MC questions on P-Bench. P-LLaVA trained with PVIT significantly outperforms other MLLMs across various question types. Remarkably, P-LLaVA demonstrates strong performances on challenging answerable tasks and unanswerable queries, where the other MLLMs drastically fail.

| MLLM | Answerable | | | | | Unanswerable | | |
| --- | --- | --- | --- | --- | --- | --- | --- | --- |
| | Crop | Aug-In | Aug-Sc-2 | Aug-Sc-3 | Avg | Adv-Img | Adv-Name | Avg |
| Qwen-VL-7B (Bai et al., 2023) | 66.58 | 72.92 | 60.87 | 50.57 | 62.74 | 1.11 | 20.62 | 10.87 |
| VILA1.5-7B (Lin et al., 2024a) | 57.54 | 48.39 | 55.91 | 38.29 | 50.03 | 36.36 | 3.23 | 19.79 |
| LLaVA-OneVision-7B (Li et al., 2024) | 84.82 | 82.17 | 77.53 | 78.31 | 80.71 | 24.12 | 1.07 | 12.60 |
| InternVL-Chat-V1.5-26B (Chen et al., 2024) | 62.99 | 56.84 | 57.14 | 51.02 | 57.00 | 53.33 | 9.64 | 31.49 |
| Deepseek-VL-7b-chat (Lu et al., 2024) | 89.72 | 24.79 | 71.43 | 66.56 | 63.13 | 4.12 | 0.00 | 2.06 |
| mPLUG-OWl2 (Ye et al., 2024) | 72.29 | 70.21 | 61.34 | 50.63 | 63.62 | 0.00 | 5.00 | 2.50 |
| P-LLaVA (Ours) | **95.42** | **97.23** | **96.48** | **97.62** | **96.69** | **99.43** | **100.00** | **99.72** |

Table 2: Evaluation of personalized description capability on P-Bench. The P-LLaVA tuned with our PVIT is able to accurately recognize and describe the person of interest, even for challenging cases where multiple individuals are contained in the image.

| MLLM | Answerable | | | | | Unanswerable | | |
| --- | --- | --- | --- | --- | --- | --- | --- | --- |
| | Cnt=1 | Cnt=2 | Cnt=3 | Cnt>=4 | Avg | Adv-Img | Adv-Name | Avg |
| Qwen-VL-7B (Bai et al., 2023) | 76.20 | 72.38 | 69.59 | 63.04 | 70.30 | 0.00 | 15.00 | 7.50 |
| VILA1.5-7B (Lin et al., 2024a) | 74.90 | 74.58 | 67.29 | 70.02 | 71.70 | 10.00 | 20.00 | 15.00 |
| LLaVA-OneVision-7B (Li et al., 2024) | 74.54 | 71.08 | 76.13 | 63.54 | 71.32 | 10.00 | 15.00 | 12.50 |
| InternVL-Chat-V1.5-26B (Chen et al., 2024) | 82.64 | 75.94 | 67.12 | 71.28 | 74.25 | 15.00 | 20.00 | 17.50 |
| Deepseek-VL-7b-chat (Lu et al., 2024) | 82.15 | 77.86 | 76.06 | 76.44 | 78.13 | 0.00 | 5.00 | 2.50 |
| mPLUG-OWl2 (Ye et al., 2024) | 80.38 | 79.11 | 77.77 | 72.68 | 77.49 | 0.00 | 5.00 | 2.50 |
| P-LLaVA (Ours) | **85.24** | **83.10** | **83.14** | **78.71** | **82.55** | **100.00** | **100.00** | **100.00** |

positive questions involves different number of people in the scene images. As the number of people grows in the scene image, it becomes more challenging for the MLLM to correctly recognize the person of interest and produce accurate descriptions.

**Evaluation Metrics** For descriptions, we perform evaluations using different strategies for answerable and unanswerable questions. For answerable ones, we adopt LongClip (Zhang et al., 2024) to evaluate the similarity between the image of the target individual and the MLLM-generated descriptions. For unanswerable ones, we calculate the percentage that the MLLM rejects to respond.

## 6 EXPERIMENTS

In this section, we demonstrate the effectiveness of our proposed PVIT on the constructed P-Bench. We first showcase the results of the PVIT-tuned LLaVA (Liu et al., 2023a), which demonstrates significantly higher performances than the SOTA MLLMs that support multi-image inputs. Then, we conduct ablation study on each of the components of our constructed data to demonstrate their contributions towards the final performance. We provide the detailed training configurations in Section G of the Appendix.

### 6.1 MAIN RESULTS ON P-BENCH

We compare the personalization capability of P-LLaVA trained with our PVIT with other SOTA MLLMs on our constructed P-Bench. We conduct evaluations using both the MC questions and personalized descriptions in Table 1 and Table 2, respectively. We observe the following phenomena for current SOTA MLLMs:

**1)** The performances of SOTA MLLMs are significantly lower with more complex inputs (i.e., Aug-In, Aug-Sc-2 and Aug-Sc-3 for MC questions, and scene images that contain more people for description questions), which indicates their limited capability and robustness in recognizing input individuals for the scene images.

**2)** All the MLLMs drastically fail for unanswerable questions. They still tend to answer the questions that are not answerable by mistakenly treating other people in the image as the person of interest. This is because the MLLMs have never been trained to reject replying to such unanswerable questions.

**3)** After fine-tuning LLaVA (Liu et al., 2023a) with our PVIT, we observe significant performance boosts for all question types in P-Bench. Specifically, we observe performance enhancement for both positive and unanswerable questions. Notably, the performance on more complex scene images is boosted even more significantly. The results verifies the effectiveness of our propose tuning strategy in improving the model's personalization capability.

## 6.2 ABLATION STUDY

Table 3: The performances on MC questions with scene images containing different numbers of people.

| MLLM | Cnt=1 | Cnt=2 | Cnt=3 | Cnt>=4 |
|---|---|---|---|---|
| Qwen-VL-7B (Bai et al., 2023) | 74.67 | 58.71 | 57.38 | 37.32 |
| VILA1.5-7B (Lin et al., 2024a) | 64.31 | 47.02 | 44.61 | 38.13 |
| LLaVA-OneVision-7B (Li et al., 2024) | 88.72 | 83.23 | 80.37 | 76.31 |
| InternVL-Chat-V1.5-26B (Chen et al., 2024) | 56.21 | 52.28 | 43.57 | 38.31 |
| Deepseek-VL-7b-chat (Lu et al., 2024) | 84.03 | 68.71 | 76.52 | 74.90 |
| mPLUG-OWl2 (Ye et al., 2024) | 78.90 | 63.23 | 77.43 | 60.51 |
| P-LLaVA (Ours) | **98.71** | **95.03** | **94.90** | **95.32** |

Table 4: Face augmentation boosts the MLLM's capabilities to recoginize individuals, while adversarial samples are critical for enabling MLLM to reject answering the unanswerable questions.

| Data | Crop | Augment | Unanswerable |
|---|---|---|---|
| Full | **96.43** | **95.32** | **99.78** |
| wo Aug | 94.93 | 88.43 | 98.73 |
| wo Adv | 95.92 | 92.48 | 1.12 |

### 6.2.1 PERFORMANCE FOR VAIROUS NUMBER OF PERSON

In Table 3, we demonstrate the MLLMs' personalization performances on MC questions with scene images containing different numbers of people. The SOTA MLLMs showcase deteriorated performances on scene images containing more people due to the challenge of accurately recognizing the specific individual of interest. On the other hand, our trained P-LLaVA remains highly accurate on such challenging cases, which verifies its capability to accurately recognize person of interest.

### 6.2.2 DATA ABLATION

We examine the effectiveness of each data component in our curated dataset in Table 4. We observe the following: 1) Face augmentation using PhotoMaker increases the diversity of input individuals, which effectively boosts the MLLM's capabilities to recoginize individuals; 2) Adversarial samples are critical for enabling MLLM to reject answering queries that are unanswerable. Without adversarial samples, the accuracy for rejecting answering unanswerable questions rapidly drops to near zero.

### 6.2.3 IMPACT OF DATA SCALE AND NAME REPETITION

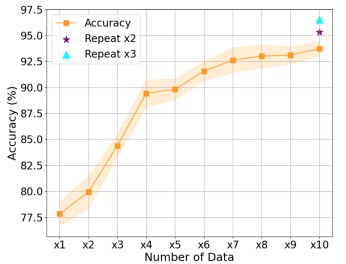

In the figure on the right, we illustrate the evaluation accuracy after training with various amounts of data. Specifically, the horizontal axis indicates the number of data units, and each unit contains 8000 samples. We observe clear performance boost when scaling up the training dataset. Furthermore, we find that even with the same data templates, by repeatedly constructing data using different names, the performance of MLLM is able to be further enhanced, which verifies that the diversity of names used during training makes the personalization capability more robust and generalizable to new individuals.

## 7 CONCLUSION

In this work, we introduce Personalized Visual Instruction Tuning (PVIT), a novel formulation a training framework that enables personalized conversations targeting specific individuals. To achieve this, We first develop an automatic pipeline that generates training data with individuals in diverse visual and conversational contexts. Then, we adopt the data to finetune the MLLM, which significantly improves MLLM's personalized dialogue capabilities, as demonstrated through the P-Bench benchmark. We hope that our work will promote the advancement in personalized applications, such as visual assistants and domestic robots, enhancing user-centric multimodal interactions.

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

## A    ETHICS CLARIFICATION

We would like to clarify that all images used in the construction of our dataset and benchmark were collected from publicly available datasets, including Visual Genome (Krishna et al., 2016), COCO (Lin et al., 2015), Object365 (Shao et al., 2019), and Flickr30k (Plummer et al., 2016). These datasets are all licensed under the Creative Commons Attribution 4.0 (CC BY 4.0) license. Additionally, all textual information (e.g., names, hobbies, and character traits) was synthesized by a large language model (LLM) through prompting and randomly paired with the images of individuals. As a result, all the textual information is fictional and does not contain any real personal information of the individuals in the images, as there are no actual correspondences between the images and the textual information. Therefore, our curated dataset does not involve any leakage of personal information. We emphasize that the goal of our dataset is to enable the multimodal large language model (MLLM) to recognize and refer to individual information within its context, rather than to memorize personal details of specific people.

## B    LIMITATION

In our work, we propose a novel in-context learning paradigm for the personalization of MLLMs, which equips the model with personalization ability using a data-centric approach, where we meticulously design an automatic pipeline for creating personalized conversation data. However, in our work, we did not take into account the bottleneck of the vision encoder's ability for producing accurate personalized representations. We assume this is one of the causes for the gap between our trained model with the performance of GPT4o. This is a parallel direction to our proposed method, which we leave as future work to further explore.

## C    DETAILS OF PVIT-3M

We elaborate the detailed statistics of our constructed PVIT-3M in Figure 3. In addition, we provide explanations of different categories in Table C. The images used in our data construction are all collected from public datasets, including Visual Genome (Krishna et al., 2016), COCO (Lin et al., 2015), Object365 (Shao et al., 2019) and flickr30k (Plummer et al., 2016).

## D    MODEL CHOICES

Throughout our paper, we used LLaVA-1.6-7B (Liu et al., 2023a) as our base MLLM for P-LLaVA. For our proposed personalized data construction pipeline, we adopt InternVL2-26B (Chen et al., 2024) for dual level textual information extraction. For textual information fusion and QA generation, we adopt LLaMA3.1-8B-instruct (Touvron et al., 2023). To crop out the people from images, we adopt GrounndingDino (Liu et al., 2023b).

## E    DETAILS OF P-BENCH

**Benchmark Data Curation**    Directly annotating the data manually introduces tedious human labour. To boost the annotation efficiency, we adopt an LLM-assisted annotation pipeline: we follow a similar annotation procedure as the one adopted in LLaMA3 (Meta, 2024). Specifically, we first feed the images containing people to GPT4o, and instruct it to design QAs for each individual. Then, we manually check the validity of the designed QAs, and match them to the corresponding individuals.

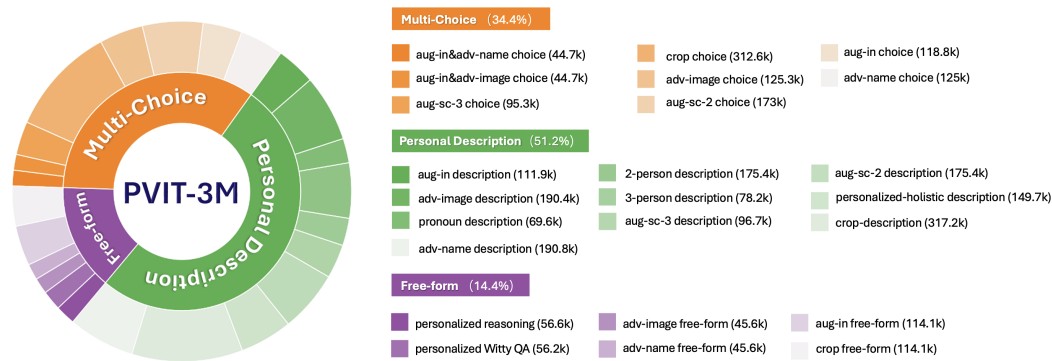

Figure 3: **Statistics of PVIT-3M**, a large scale personalized instruct tuning dataset. Left: Data Distribution within Each Category. The outer circle shows the distribution of all data categories and the inner circle shows the distribution of data subsets. Right: The detailed quantities of datasets.

| Keyword | Explanation |
|---|---|
| Crop | Uses cropped faces in the personalized prefix. |
| Aug-In | Uses generated photos based on the cropped faces in the personalized prefix. |
| Aug-Sc-x | Involves concatenating different cropped individuals into one image to replace the original scene image. The number "x" following "Aug-Sc" indicates the number of individuals concatenated. |
| Adv-Name | Indicates an unanswerable scenario where the name in the prefix is different from the name mentioned in the QA. |
| Adv-Image | Indicates an unanswerable scenario where the individual photo in the prefix is not present in the scene image. |
| Pronoun Description | Replaces specific names with pronouns in the description. |
| x-person description | Asks the MLLM to give a description of "x" individuals while providing augmented scene image that concats "x" persons. |
| Personalized-Holistic Description | The query is changed to a general one, such as "describe this image", while requiring the model to include certain individual's name from the prefix in the overall description of the image. |
| Personalized Reasoning | Asks reasoning-based questions involving the specific individual. |
| Personalized Witty QA | Generates humorous and creative responses involving the specific individual. |

Table 5: Explanation of PVIT-3M Dataset Categories.

**Benchmark Statistics** P-Bench contains two main categories: Multiple Choice (MC) and Description. The statistics for each category are detailed below, along with the total counts.

MULTIPLE CHOICE (MC) CATEGORY

- **Type Statistics:** The MC category includes six types of samples, totaling 915 samples:
    - Aug-Sc-2: 100 samples
    - Aug-Sc-3: 100 samples
    - Adv-Image: 100 samples
    - Adv-Name: 100 samples
    - Aug: 100 samples
    - Crop: 415 samples

- **Number of People in Scene Image:** Total of 415 images:
    - Images containing 1 person: 221
    - Images containing 2 people: 85
    - Images containing 3 people: 48
    - Images containing 4 or more people: 61

DESCRIPTION CATEGORY

- **Type Statistics:** The Description category includes a total of 100 samples, divided as follows:
    - **Answerable (Aug-In):** 60 samples
    - **Unanswerable:** 40 samples, consisting of:
        * Adv-Image: 20 samples
        * Adv-Name: 20 samples
- **Number of People in Scene Image:** Total of 60 images:
    - Images containing 1 person: 14
    - Images containing 2 people: 31
    - Images containing 3 people: 6
    - Images containing 4 or more people: 9

---

**Holistic Information Generation Prompt**

<image>
Provide a detailed description of this image, with special emphasis on the main character, including their appearance, expressions, actions, and any distinguishing features.

---

Table 6: The prompt for generating *holistic information*.

---

**Personal Information Generation Prompt**

<image>
Describe the person in this image. Focus on this person's main features. Remember, **Do Not** include any background information. Additionally, in your response, you should use <name> to refer to the person you describe when you mention the person's name first time. Again, you must contain <name> in your response.

---

Table 7: The prompt for generating specific *person information*. <image> represents the cropped photo of individual from the scene image. Here, we ask the MLLM to use the placeholder <name> to refer the individual when describing.

## F    DETAILED PROMPTS FOR DATA GENERATION

We provide the detailed prompts for each phase of data generation. In Table 6 and Table 7, we showcase the prompts provided to the MLLM, which are used for generating the holistic information and the personal information, respectively. In Table 8, we demonstrate the prompt used for integrating the dual-level information into a single description, which is provided to the LLM along with the holistic and personalized information. In Table 9, we demonstrate the prompt to LLM that is used for creating the Multi-Choice QA. For dual-level information fusion and Multi-Choice QA generation, we incorporate manually designed in-context examples to guide the LLM.

---

**Dual-level Information Fusion Prompt**

---

**# TASK DESCRIPTION**

Given the Person Information of a person and the Holistic Information of the whole image, you need to put the placeholder <name >in the Holistic Information to represent the person described in Person Information. You should reply starting with "Combined Description:".

**# EXAMPLE**

**Person Information**: *<name> is sitting in a relaxed posture. <name> is wearing a dark blue T-shirt and light blue jeans. On his left wrist, he has a watch. <name> is smiling and appears to be in a jovial mood. He is holding a blue object in his right hand, which looks like a piece of cloth or a towel. The background is a simple blue wall, and there is a light-colored blanket or couch behind him.*

**Holistic Information**: *The image captures a heartwarming moment between a man and a young boy. The man, wearing glasses and a blue shirt, is sitting on a couch, holding a blue balloon in his hand. He is smiling and looking at the boy, who is seated next to him. The boy, wearing a green shirt, is also smiling and looking at the man. The background of the image features a blue wall, adding to the overall warmth of the scene. The man's position on the left and the boy's on the right create a balanced composition. The blue balloon held by the man adds a playful element to the image. The smiles on their faces suggest a moment of joy and connection between the two.*

**Integrated Information**: *The image captures a heartwarming moment between <name> and a young boy. <name>, wearing glasses and a blue shirt, is sitting on a couch, holding a blue balloon in his hand. He is smiling and looking at the boy, who is seated next to him. The boy, wearing a green shirt, is also smiling and looking at <name>. The background of the image features a blue wall, adding to the overall warmth of the scene. <name>'s position on the left and the boy's on the right create a balanced composition. The blue balloon held by <name> adds a playful element to the image. The smiles on their faces suggest a moment of joy and connection between the two.*

**# TASK**
Person Information: {Person Information}
Holistic Information: {Holistic Information}
Integrated Information:

Table 8: The prompt for fusing person information and holistic information. The {Person Information} and {Holistic Information} are placeholders that will be replaced by the actual content to be processed.

## G  TRAINING DETAILS

In Table 10, we illustrate the detailed hyper-parameters used when fine-tuning the MLLM with our PVIT-3M. We wish to note that we start tuning from the checkpoint of LLaVA-7B Liu et al. (2023a). We train the MLLM with a subset of our PVIT-3M with 1M samples. The entire training is conducted on 8 A100 GPUs with 80GB memory, which lasted for 30 hours.

## H  PERSONALIZED ATTRIBUTE CREATION

To create personal attributes, we adopt a strategy similar to name swapping as described in Section 4.3.1. Specifically, we first predefine a set of categories for attributes. In our work, we mainly consider the following: birthday, hobbies, family, education, favorite things and personality. Then, for each category, we prompt the LLM to generate a list of attributes. The prompts are provided in Table 11. These attributes are dynamically combined with the person's images and names when constructing the QA pairs. We show the results of trained MLLM in Figure 4, which demonstrates that our model can correctly associate the personal attributes with their photos.

---

**Multi-Choice QA Generation Prompt**

**# TASK DESCRIPTION**

Now you need to generate multiple-choice questions based on Information. You should pay particular attention to the characteristics mentioned in the description that describe this person, and use these characteristics to create questions and possible answers.

**# RESPONSE FORMAT**

Your response must strictly follow the format below:
[["question": "...", "choices": ["...", "...", "...", "..."], "answer": "..."]]

**# ATTENTION**

1. Please ensure that all references to the person in your questions and answers are replaced with the placeholder <name >.
2. Only generate multiple-choice questions about the individual.
3. Ensure that each set of choices has clear distinctions and no overlap to avoid multiple correct answers.

**# EXAMPLE**

**Information**: *In the photo, <name> is wearing a white shirt and blue jeans. She is standing beside a man in a blue T-shirt and has her hands on her hips. She is also wearing a black bag.*

**Generated MC**: *["question": "What color shirt is <name> wearing?", "choices": ["Red", "White", "Blue", "Black"], "answer": "White"], ["question": "What color are <name>'s jeans?", "choices": ["Black", "Green", "Blue", "Yellow"], "answer": "Blue"], ["question": "What is <name> doing with her hands?", "choices": ["Holding a bag", "Hands on her hips", "Waving", "In her pockets"], "answer": "Hands on her hips"], ["question": "What accessory is <name> wearing?", "choices": ["A hat", "A scarf", "A black bag", "Sunglasses"], "answer": "A black bag"]*

**#TASK**

Information: {Information}

Generated MC:

---

Table 9: Prompt for generating Multi-Choice QA.

## I  RELATIONSHIP WITH VISUAL PROMPTING

Recently, visual prompting has garnered a lot of attention. Set-of-marks (Yang et al., 2023) first observed that after associating each object with a mask and a tag, GPT4-V is able to associate each region according to the user input. Inspired by this finding, Draw-And-Understand (Lin et al., 2024b) designs a visual prompt encoder and that can encode various visual prompts (points, bounding boxes, and free-form shape), which helps the MLLM better understand fine-grained features in the image. This line of work also has the potential to improve MLLM's personalized capability, which is in parallel with our proposed in-context learning paradigm and can be applied in conjunction: the former enhances fine-grained understanding of various objects in the same image, while the latter helps understanding the relationship between different images (e.g., personal image and scene image).

## J  FAILURE CASES

In Figure 5, we demonstrate the failure cases of our P-LLaVA. The leftmost and middle figures demonstrate P-LLaVA's capability to recognize a person of interest at various ages. They show that the model performs well when the age difference is relatively small but struggles to identify the individual as the age gap widens. The rightmost figure reveals another limitation: P-LLaVA

Table 10: Hyper-parameters used for PVIT.

| Parameter | Value |
|---|---|
| --lora_enable | True |
| --lora_r | 128 |
| --lora_alpha | 256 |
| --mm_projector_lr | 1e-4 |
| --deepspeed | ./scripts/zero2.json |
| --version | v1 |
| --vision_tower | openai/clip-vit-large-patch14-336 |
| --mm_projector_type | mlp2x_gelu |
| --mm_vision_select_layer | -2 |
| --mm_use_im_start_end | False |
| --mm_use_im_patch_token | False |
| --image_aspect_ratio | pad |
| --group_by_modality_length | True |
| --bf16 | True |
| --num_train_epochs | 1 |
| --per_device_train_batch_size | 16 |
| --per_device_eval_batch_size | 4 |
| --gradient_accumulation_steps | 2 |
| --evaluation_strategy | no |
| --save_strategy | steps |
| --save_steps | 50000 |
| --save_total_limit | 1 |
| --learning_rate | 2e-4 |
| --weight_decay | 0. |
| --warmup_ratio | 0.03 |
| --lr_scheduler_type | cosine |
| --logging_steps | 1 |
| --tf32 | True |
| --model_max_length | 4096 |
| --gradient_checkpointing | True |
| --dataloader_num_workers | 4 |
| --lazy_preprocess | True |

---

**Attribute Generation Prompt**

You are tasked with creating detailed and unique attributes for the general category of attributes: <category>. For this category:
1. Break it down into diverse and non-overlapping attributes or aspects.
2. Ensure the attributes are well-defined, specific, and meaningful.

---

Table 11: The prompt for generating a list of attributes for a category. <category> represents the a category of attributes, such as birthday, hobbies, and personality.

has difficulty distinguishing the person of interest from others who closely resemble them, such as impersonators present in the scene. These failures could stem from challenges that are inherently difficult, even for humans—for instance, when the age gap is substantial or when the impersonator or a sibling bears a strong resemblance to the person of interest.

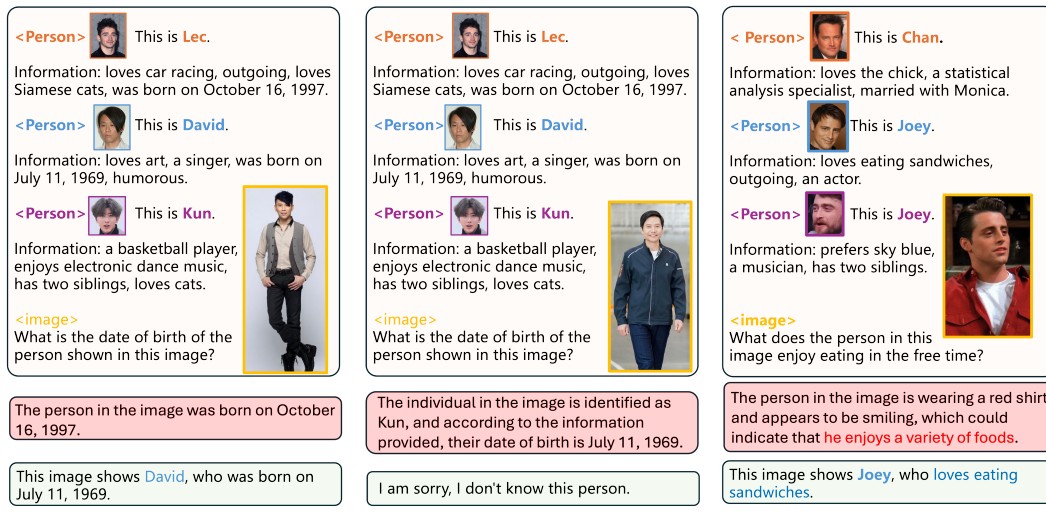

Figure 4: P-LLaVA's ability to associate the person of interest with his/her attributes. Left: P-LLaVA successfully recognizes the person in the scene image with his attributes (birthday). Middle: P-LLaVA rejects to respond when the person's information is not provided to it. Right: P-LLaVA can correctly identify the person of interest even if there are multiple people with the same name. The red square and green square contain the responses from the original MLLM and P-LLaVA, respectively.

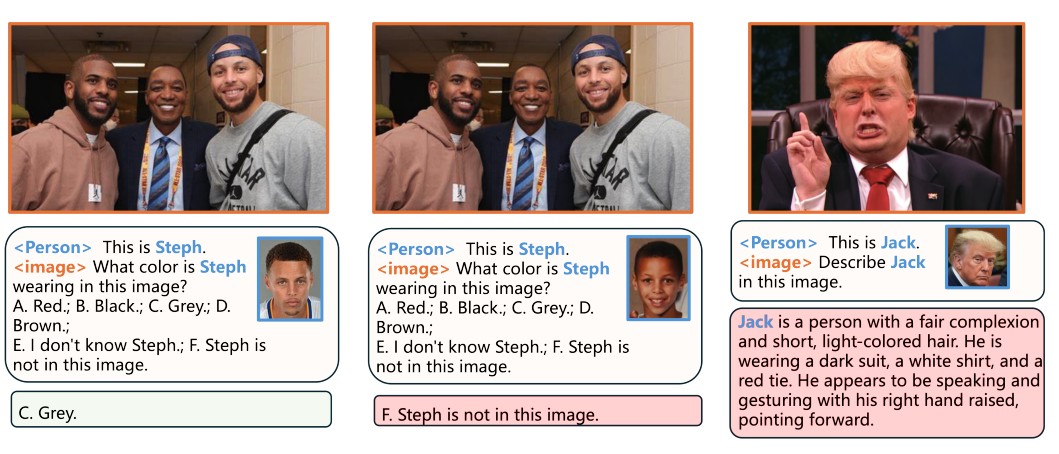

Figure 5: The leftmost and middle figures illustrate P-LLaVA's ability to recognize a person of interest at different ages. These figures indicate that the model can successfully identify the person of interest when the age gap is relatively small. However, when the age gap becomes significant, the model fails to recognize the individual. The rightmost figure highlights another limitation: P-LLaVA struggles to differentiate between the person of interest and individuals who closely resemble them, such as impersonators in the scene.

