# OpenReview forum: "Personalized Visual Instruction Tuning"
_ICLR.cc/2025/Conference — ICLR 2025 Poster_

### Official Review · Reviewer_ZV1c · 2024-10-27

**Soundness:** 2
**Presentation:** 2
**Contribution:** 2
**Rating:** 6
**Confidence:** 5

**Summary:**

This paper tries to tackle the "face blindness" issue of the multimodal large language models. While, it is quite questionable whether you need the model to handle "face blindness" directly, or because it is safety and ethics issues, the model needs some mechanism to handle this issue safely. This is an important question for the motivation of this paper.

Technically, this paper presents a method to collect high-quality data for personalized conversations, then proposes Personalized
Visual Instruction Tuning (PVIT) to enable the MLLMs (LLaVA in the paper) for personalized dialogues. This paper also constructs a P-Bench for evaluation purpose. Empirical experiments on P-Bench show that the P-LLaVA can do well in the personalized setting.

**Strengths:**

Technical Strengths:

1. This paper proposes a way to curate the personalized instruction conversations, and also introduces a P-Bench for evaluation purpose.

2. It finetuned LLaVA in the personalized setting, and demonstrates proposing results.

**Weaknesses:**

1. One major issue is there might lack a section to discuss the potential Ethics issues since it involves people in the dataset and models.

2. Methodologically, the data curation is okay, the model training part is incremental.

**Questions:**

1. As suggested, one potential major issue is about the ethics issues about the people topic, which needs more discuss. And also need safety evaluation for the data, benchmark and models.

2. One question is - after you adapt the LLaVA model to the personalized setting, how its (P-LLaVA) performance on the original general benchmarks.

3. And one more question is - just curious, how is the performance of GPT-4o or GPT-4v, Gemini or these frontier models on the proposed P-Bench? How they handle this people/face issue.

**Details Of Ethics Concerns:**

This needs a section to discuss the potential ethics issues, both the dataset and models. Person (or even celebrity), even GPT, sometimes, they response with "I do not know" or "policy issue" etc.

Note: my current score is a temporary score, only evaluate this paper from a technical perspective. Final score may depend on how people feel the ethics issues for this work, especially, its motivation is "face blindness".

---

> ### Author Response · Authors · 2024-11-22
> **Response to Reviewer ZV1c**
>
> Thank you for your insightful advice and constructive comments, we appreciate that you find our proposed data construction framework and benchmark useful for the community. We aim to address your concerns in the following:
>
>
> **Potential Ethics Issues**: Thank you for pointing ourt this potential confusion. We would like to clarify this from the following aspects:
>
> - Firstly, we would like to clarify that all images used in our data construction were collected from publicly available datasets, including Visual Genome, COCO, Object365, and Flickr30k. Additionally, all textual information (e.g., names, hobbies, and character traits) was synthesized by a large language model (LLM) through prompting and randomly paired with the images of individuals. Consequently, there are no true correspondences between the images and the textual information. As a result, our curated dataset does not involve any leakage of personal information.
>
> - Secondly, since the personal text information is fully synthesized by the LLM, and randomly paired with personal images during training, the MLLM is not to trained to memorize any specific individual, rather, it is trained to recognize target individual in the image when it is provided in the MLLM’s context. This training approach ensures that no privacy risks are introduced.
>
> - In addition, we will make sure the released dataset uses the same license as the source datasets (CC BY 4.0).
>
> We’ve added this paragraph to Section I of the Appendix for clarification.
>
> **Novelty of Our Method**: We are truly grateful for your acknowledgement of our data generation framework. We wish to clarify that we are the first to formulate personalization of MLLM as an in-context learning problem, which only needs to train the MLLM once, and requires no additional training for each newly introduced individual. To achieve this, we take a data-centric approach: we design a novel data construction framework that automatically creates personalized conversation data without requiring human labelling. Compared with previous model-centric approaches that tackle this task by modifying network architectures [1,2], our in-context learning paradigm and data-driven method is more generalizable and practical.
>
> We also point out that the research community today prioritizes the development of generalist models rather than task-specific ones. Consequently, taking a data-centric perspective, minimizing modifications to model architecture and training paradigms is advantageous, as it allows training to remain unified across different tasks.
>
> **Performance on Original General Benchmarks**: Great question! If we only finetue the MLLM with personalized training data, it is inevitable that the MLLM will suffer from catastrophic forgetting. Therefore, during our training, we’ve included a subset (300k) randomly sampled from the original training set of LLaVA-Instruct-665k to alleviate forgetting. We observe a slight or no decrease in performances on the regular benchmarks compared with original LLaVA 1.6.
>
> If computational resource allows, we could incorporate more conventional conversations in our training data, which we expect would preserve the original capability,
>
> | | VQAv2 | GQA | MME | PoPE (lower is better)  |
> |-|-|-|-|-|
> | LLaVA-1.6-7B| 81.83|63.28|1376/327| 0.86/0.88/0.87 |
> | P-LLaVA-1.6-7B| 79.08|62.93|1345/344| 0.87/0.88/0.87 |
>
>
> **How is the performance of GPT-4o**: We’ve conducted additional experiments on GPT-4o using both API calls and chat windows from the webpage. We find that the API calls to GPT4o often refuse to answer personalized questions and simply respond with “I don’t know”, while the chat window from the webpage functions quite well. Therefore, we randomly select 30 samples and test each of them manually in the webpage’s chat window, and find that all the questions are correctly answered. We presume that GPT4-o, as a closed sourced model, may have incorporated similar data during training, which empowers it with personalization capability. We believe our proposed data construction framework and curated dataset helps close the gap between GPT4-o and open-source MLLMs on personalized conversations.
>
> [1] MyVLM: Personalizing VLMs for User-Specific Queries
>
> [2] Yo'LLaVA: Your Personalized Language and Vision Assistant

---

> ### Author Response · Authors · 2024-11-24
> **Looking forward to further discussion**
>
> Dear Reviewer ZV1c,
>
> We deeply appreciate your insightful suggestions and helpful comments. It is an honor to know that you find our work proposes a practical framework for personalization data synthesis, curates a useful benchmark for the community, and presents good experimental results.
>
> Your help in identifying potential concerns in our manuscript has been invaluable, and we have diligently addressed these in our response. Since the discussion period deadline is approaching, we would greatly appreciate any further comments you may have on our response. We are eager to resolve any additional concerns you might have.
>
> We fully understand your busy schedule and are genuinely thankful for the time you have dedicated to helping us enhance our work. We look forward to receiving any additional feedback you may offer.

---

> > ### Comment · Reviewer_ZV1c · 2024-11-27
> >
> > Thanks for the response.

---

### Official Review · Reviewer_C8KK · 2024-11-03

**Soundness:** 3
**Presentation:** 3
**Contribution:** 3
**Rating:** 8
**Confidence:** 4

**Summary:**

This paper introduces Personalized Visual Instruction Tuning (PVIT), a novel framework that enables multimodal large language models (MLLMs) to engage in personalized, coherent dialogues tailored to specific individuals within images. The PVIT framework addresses the limitations of existing MLLMs, which often fail to recognize individuals and deliver contextually accurate, personalized responses. The paper proposes a comprehensive pipeline for personalized multimodal data construction. The experiments demonstrate that models fine-tuned with PVIT show substantial improvements in personalization tasks.

**Strengths:**

1. PVIT’s multi-phase pipeline is designed to generate personalized training data. By integrating visual expert models, image generation tools, and large language models, the authors ensure a rich and diverse dataset that enhances the personalization capabilities of MLLMs.

2. The authors contribute P-Bench, a novel benchmark designed to evaluate personalization capabilities in MLLMs.

3. Experiments demonstrate that PVIT improves MLLMs’ accuracy in handling personalized queries across various tasks. The ablation studies further validate the impact of component designs in the pipeline, highlighting the framework’s effectiveness.

**Weaknesses:**

1. Reliance on Pretrained MLLMs and Potential Hallucinations: The pipeline relies heavily on pretrained multimodal large language models (MLLMs) for tasks such as image captioning and textual information generation. However, these models are known to sometimes produce hallucinated details. Without mechanisms to detect or filter out such inaccuracies, the generated data could introduce noise into the training process, leading to degraded model performance or unpredictable responses. To strengthen the pipeline's reliability, the authors should provide examples of failed cases and outline a strategy for identifying and filtering out unreasonable or erroneous data.

2. Ethical and Privacy Concerns in Human Image Sourcing: The paper does not mention the origin or sourcing of human images used for training and testing, which is critical from both ethical and privacy standpoints. Without a clear explanation of how these images were obtained and the measures taken to ensure consent and privacy, the approach may risk ethical violations. The authors should explain more about the original metadata collection.

3. The limitation section, which is required, is missing.

**Questions:**

1. Which version of LLaVA is finetuned as P-LLaVA?

2. What models are used in dual-level textual information extraction and fusion?

**Details Of Ethics Concerns:**

The origin or sourcing of human images used for training and testing should be claimed in this work.

---

> ### Author Response · Authors · 2024-11-22
> **Response to Reviewer C8KK**
>
> Thank you for your insightful advice and constructive comments, we appreciate that you find the proposed multi-phase data construction pipeline produces rich and diverse personalized training data, curates a novel benchmark, and achieves impressive experimental results. We aim to address your concerns in the following:
>
> **Reliance on Pretrained MLLMs and Potential Hallucinations**: Great point! For holistic information of the entire image, we prompt the MLLM (InternVL2-26B) to describe the overall content of the image that captures the main elements. For personal information, we first crop out each individual in the image, and let the MLLM only describe that individual. With the above prompts, we observe that the MLLM produces compact descriptions that are not prone to hallucination. To verify this, we evaluate our P-LLaVA on the hallucination benchmark PoPE [3], which shows similar performance as the original LLaVA 1.6:
> || Adversarial | Random | Popular |
> |-|-|-|-|
> | LLaVA-1.6-7B| 0.86 | 0.88 | 0.87 |
> | P-LLaVA-1.6-7B|  0.87 | 0.88 | 0.87 |
>
> The results demonstrate that after personalized training, the MLLM does not become more prone to hallucination.
>
> However, we agree that hallucination may still potentially occur. Therefore, as inspired by image textualization [1], we first extract the objects contained in each description, and then employ an open vocabulary detector GroundingDino [2] to check for the existence of each object mentioned in the description. We then keep only the descriptions that do not contain hallucinated objects for subsequent stages.
>
> **Ethical and Privacy Concerns in Human Image Sourcing**: Thank you for pointing ourt this potential confusion. We would like to clarify this from the following aspects:
>
> - Firstly, we would like to clarify that all images used in our data construction were collected from publicly available datasets, including Visual Genome, COCO, Object365, and Flickr30k. Additionally, all textual information (e.g., names, hobbies, and character traits) was synthesized by a large language model (LLM) through prompting and randomly paired with the images of individuals. Consequently, there are no true correspondences between the images and the textual information. As a result, our curated dataset does not involve any leakage of personal information.
>
> - Secondly, since the personal text information is fully synthesized by the LLM, and randomly paired with personal images during training, the MLLM is not to trained to memorize any specific individual, rather, it is trained to recognize target individual in the image when it is provided in the MLLM’s context. This training approach ensures that no privacy risks are introduced.
>
> - In addition, we will make sure the released dataset uses the same license as the source datasets (CC BY 4.0).
>
> We’ve added this paragraph to Section I of the Appendix for clarification.
>
> **Limitation Section**: Thank you for the reminder. Here are the limitations of our current work: In our work, we propose a novel in-context learning paradigm for the personalization of MLLMs, which equips the model with personalization ability using a data-centric approach, where we meticulously design an automatic pipeline for creating personalized conversation data. However, in our work, we did not take into account the bottleneck of the vision encoder’s ability for producing accurate personalized representations. We assume this may be one of the cause for failing to differentiate two people that look very similar (Figure 5 in the Appendix). This is a parallel direction to our proposed method, which we leave as future work to further explore. We’ve added the limitation section to our updated paper.
>
> **LLaVA Version, Models used in information extraction and fusion**: Thank you for pointing it out. We used LLaVA-1.6 as our base MLLM throughout our paper. For textual information extraction, we adopt InternVL2-26B. For information fusion and QA generation, we adopt LLaMA3.1-8B-instruct. We’ve added these details in section C of the Appendix.
>
> [1] Image Textualization: An Automatic Framework for Creating Accurate and Detailed Image Descriptions
>
> [2] Grounding DINO: Marrying DINO with Grounded Pre-Training for Open-Set Object Detection
>
> [3] Evaluating Object Hallucination in Large Vision-Language Models

---

> > ### Comment · Reviewer_C8KK · 2024-11-23
> >
> > Thanks for the authors' reply. I believe the released data is useful for future research in the related community. Looking forward to seeing the open-source dataset and dataset construction pipelines. I have updated my score.

---

> > > ### Author Response · Authors · 2024-11-24
> > > **Response to Reviewer C8KK**
> > >
> > > Thank you very much for your acknowledgement on the significance of our data generation framework and curated dataset! We appreciate the time and effort you dedicated in reviewing our paper. Your suggestions and comments have been invaluable in improving our manuscript!

---

### Official Review · Reviewer_dSk8 · 2024-11-03

**Soundness:** 3
**Presentation:** 3
**Contribution:** 2
**Rating:** 6
**Confidence:** 4

**Summary:**

This paper introduces Personalized Visual Instruction Tuning (PVIT), a framework that enables MLLM to engage in personalized conversations with specific individuals. PVIT addresses the “face blindness” limitation of MLLMs by training them to recognize individuals within images and generate contextually accurate responses.

NOTE:

Post-rebuttal: After the first rebuttal round, I have decided to raise my score from 5 to 6 since the author's responses have cleared my most concerns.

**Strengths:**

1. PVIT presents a novel approach to MLLM personalization by leveraging in-context learning capabilities, avoiding the need for additional training for each individual.
2. A benchmark is proposed for future research on this direction.
3. The paper is well-written and clearly explains the PVIT framework, data generation process, and experimental setup.

**Weaknesses:**

1. Some typos like Line 191-192 Two special tokens are mentioned, however, only one is described
2. The current implementation focuses primarily on names and faces.
3. The paper primarily focuses on individuals present in the training data. Some investigation on the zero-shot ability would help to understand this framework's ability.
4. This problem is an interesting setting, but the proposed methods is not that novel.
5. Current experiments are not that enough, more experiments as described in questions are welcome.

**Questions:**

1.  Line 277: More advanced knowledge, such as character, hobbies, and professions can be easily
extended and will be considered in future work. -> About How? Like character would be quite abstract to describe.
2. How about the temporal changes, like inputting a child's image while asking for the same person with quite different ages?
3. How about the twins with similar names and similar faces?
4. What happened to the situations where two different with similar faces yet different names or similar names yet different faces?

---

> ### Author Response · Authors · 2024-11-22
> **Response to Reviewer dSk8**
>
> Thank you for your helpful suggestions and insightful comments, we appreciate that you find we propose a novel approach for personalization, curates a helpful benchmark, and paper is well written. We aim to address your concerns in the following:
>
> **Typos**: Thank you for pointing it out. We fixed the typo in our current version.
>
> **Current Implementation Focus on faces**: Thank you for your insightful comment. We would like to emphasize that we did not claim to address general personalization, despite our framework being inherently adaptable to various objects by design. As highlighted in our abstract, our work specifically targets “face blindness.” We chose to focus on humans because personalization in this context has the most widespread application.
>
> However, our data construction framework is designed to be generalizable to accommodate various types of subjects beyond humans. For instance, during object extraction, an open-vocabulary detector can be employed to identify other objects (e.g., dogs), instead of persons. We wish to note that personalization for other objects often depends on specific user scenarios and can be adapted to their unique needs. For example, a vehicle manufacturing company may have requirements for vehicle personalization and should be able to supply abundant domain-specific training data, based on which our framework can create personalized training data in that domain.
>
> Therefore, rather than training a universal MLLM capable of personalizing for all types of objects—a goal that is infeasible since we can not exhaustively capture all possible categories—we aim to provide a customizable framework that can be tailored to different use cases.
>
>
> **Zero-shot ability**: We would like to clarify that our in context learning paradigm indeed emphasizes zero-shot capability during inference, which allows the MLLM to generalize to new individuals of interest without further training. We wish to note that the data in our benchmark is strictly outside of the training data.
>
> **Novelty of Our Method**: Thank you for the acknowledgement. We wish to clarify that we are the first to formulate personalization of MLLM as an in-context learning problem, which only needs to train the MLLM once, and requires no additional training for newly introduced individual. To achieve this, we take a data-centric approach: we design a novel data construction framework that automatically creates personalized conversation data without human labelling. Compared with previous model-centric approaches that tackle this task by modifying network architectures [1,2], our in-context learning paradigm and data-driven method is more generalizable and practical.
>
> **More advanced knowledge**: We mainly focused on faces and names due to the following reason: the major challenge of our in context learning formulation of personalization requires accurately recognizing the target individual in scene images, rather than comprehending text-based attributes. The former is a new ability not possessed by SOTA open source MLLMs, while the latter is easier to achieve due to the strong ability of the base LLM.
>
> As suggested, we conduct addition experiments using more advanced personal attributes. To create such data, we first define several categories like *hobbies, profession and character*. Then, each category of attributes can be created in a similar manner as name: we prompt the LLM to create a list of each attribute, and dynamically combine them with the face image when constructing the QA data.
>
> During the rebuttal period, we attempted including those new attributes and conducted training, and demonstrated the model output after fine tuning in Section G, Figure 4 of the appendix. We observe that the MLLM is able to associated the attributes with the corresponding target individual. We also provide the prompts for synthesizing these attributes, which can be extended to other types of attributes.
>
> **Temporal changes, twins with similar names and similar faces, similar faces yet different names or similar names yet different faces**: Great point! We tried some examples as you suggested, and demonstrate the results in Figure 4 and Figure 5 of the appendix.
> - We find that the MLLM can successfully recognize the person of interest when there are multiple people with the same name (rightmost in Figure 4).
> - We find that when the age difference is not too significant, the MLLM is able to recognize the  target person. However, it fails when the age gap is too large (leftmost, middle in Figure 5).
> - We also find that the MLLM sometimes fail to differentiate the target person when there is another person that looks very similar (rightmost in Figure 5).
>
> However, we wish to note that these cases may be challenging even for humans, which requires further efforts to address.
>
> [1] MyVLM: Personalizing VLMs for User-Specific Queries [2] Yo'LLaVA: Your Personalized Language and Vision Assistant

---

### Official Review · Reviewer_YLzK · 2024-11-04

**Soundness:** 3
**Presentation:** 2
**Contribution:** 2
**Rating:** 5
**Confidence:** 4

**Summary:**

This paper tackles personalization issue in MLLM. Current MLLM struggle to recognize and engage with specific people in images, making them less useful for personal AI assistants. The authors propose Personalized Visual Instruction Tuning (PVIT), which lets MLLMs handle personalized conversations by introducing people as a multimodal prefix without needing extra training. They created a streamlined data pipeline to automatically generate large-scale, personalized datasets and a new benchmark, P-Bench, to measure personalization abilities.

**Strengths:**

1. The proposed Personalized Visual Instruction Tuning (PVIT) MLLMs conduct personalized conversation based on a general MLLM for personalized context without further tuning during test time.
2. The benchmark P-Bench of around 1500 samples could be used on relevant tasks.
3. Good qualitative and quantitative results.
4. The proposal method is easy to follow and the presentation is clear.

**Weaknesses:**

1. In other work like Yo'llava, the personalization including various types of subjects, while in this work, it is limited to human face, which is different to the claim in the paper such as in the introduction.
2. For evaluation comparisons, many actual SOTA MLLMs are not included such as GPT-4. I tried those models and found out they perform pretty good in the task. For example, for GPT4o are correct on several examples in Figure 2.
3. No failure cases analysis. I do not think such method could resolve personalization MLLM perfectly, but it lacks the detailed analysis.
4. Some proposed mechanisms are already commonly used in previous methods. For example, Personalized Wrapper Tokens basically denote the added information with special token.

**Questions:**

1. Does it work well on non-human objects?
2. For the MLLM used for comparisons, what are the settings? For example, some models are designed for one image input, how do you use it in multi-image input setting?

---

> ### Author Response · Authors · 2024-11-22
> **Response to Reviewer YLzK**
>
> Thank you for your constructive and insightful advice, we appreciate that you find our proposed method to achieve good personalization performance, and the proposed benchmark is potentially helpful for the community. We aim to address your concerns in the following:
>
> **Various types of subjects**: Thank you for your insightful comment. We wish to clarify that the primary goal of our paper is to demonstrate the effectiveness of using in-context learning for the personalization of MLLMs and to propose an automated framework for synthesizing training data.
>
> Our data construction framework is designed to be generalizable and can be extended to accommodate various types of subjects beyond humans by synthesizing corresponding training data. For instance, during object extraction, an open-vocabulary detector can be employed to identify other objects, such as dogs, instead of persons.
>
> However, rather than training a universal MLLM capable of personalizing for all types of objects—a goal that is infeasible since we can not exhaustively capture all possible object categories—we aim to provide a customizable framework that can be tailored to different use cases.
>
> We chose to focus on humans because personalization involving human subjects is one of the most prevalent and impactful real-world applications. It benefits from an abundance of publicly available, relevant data. In contrast, personalization for other objects often depends on specific user scenarios and can be adapted to their unique needs. For example, a vehicle manufacturing company might have significant requirements for vehicle personalization and should be able to supply abundant domain-specific training data for this purpose.
>
> **Comparison with GPT4-o**: We’ve conducted additional experiments on GPT-4o using both API calls and chat windows from the webpage. We find it strange that the API calls to GPT4o often refuse to answer personalized questions and simply respond with “I don’t know”, while the chat window from the webpage functions quite well.
>
> Therefore, we randomly select 30 samples and test each of them manually in the webpage’s chat window, and find that all the questions are correctly answered. We presume that GPT4-o, as a closed sourced model, may have incorporated similar data during training, which empowers it with personalization capability. We believe our proposed data construction framework and curated dataset helps close the gap between GPT4-o and open-source MLLMs on personalized conversations.
>
>
>
> **Analysis on failure case**: Indeed, our method does not achieve perfect personalization in all scenarios. Specifically, we observe the MLLM may struggle to accurately identify the target individual if the age gap is too significant, and sometimes fails to differentiate the people that look too similar (Shown in Figure 5 of the appendix). We believe even though this problem is challenging, even for humans, it can be potentially solved by incorporating more hard samples into the training data. We will further explore this direction in the future.
>
> **Personalized Wrapper Tokens**: Our approach differs from previous methods like Yo'llava. Instead of adding a unique special token and fine-tuning the model for each new concept, we use just two delimiter symbols to wrap information about an individual. This allows the model to generalize to new individuals without additional training at test time, making our method more scalable and efficient.
>
> **How do you use MLLMs designed for single inputs in multi-image input setting**: We wish to clarify that all the baseline MLLMs we selected are able to accept multiple images as input.

---

> > ### Comment · Reviewer_YLzK · 2024-11-25
> >
> > Thanks for the rebuttal!
> >
> > 1. As in my initial review, only results on human faces can not support the claim in the paper. To demonstrate the concept of "personalization" rather than "human face personalization", it is crucial to include other types of objects.
> > 2. Thanks for the exp on GPT-4o. I got the same results with web version. This conclusion influenced many finding in the paper such as "The results indicate that the ability of current SOTA MLLMs to perceive personalized concepts is
> > limited, which can be significantly boosted after training with our proposed PVIT", which firstly undermine the impact of this paper and then reduce the correctness of some main claims in this paper.
> > 3. Thanks for the analysis. I believe a more detailed one is important to support the claim in this paper.
> > 4. I did not mean the special token that needs to be further trained for each concept. I was also talking about those general token. For example, when adapting GPT-2 to a chat version, or adapt llm such as llama to llava, people commonly use some tokens like <image> </image>, <conv>, so to incorporate new information. What are the difference from yours and those common method?

---

> ### Author Response · Authors · 2024-11-24
> **Looking forward to further discussion**
>
> Dear Reviewer YLzK,
>
> We deeply appreciate your insightful suggestions and helpful comments. It is an honor to know that you find our work proposes a practical paradigm for personalization, curates a useful benchmark for the community, and presents good qualitative and quantitative results.
>
> Your help in identifying potential concerns in our manuscript has been invaluable, and we have diligently addressed these in our response. Since the discussion period deadline is approaching, we would greatly appreciate any further comments you may have on our response. We are eager to resolve any additional concerns you might have.
>
> We fully understand your busy schedule and are genuinely thankful for the time you have dedicated to helping us enhance our work. We look forward to receiving any additional feedback you may offer.

---

> ### Author Response · Authors · 2024-11-25
> **Response to Reviewer YLzK**
>
> Thank you for your response,  we further clarify for each of your newly raised questions below:
>
> - ***Scope and Application***: **We would like to emphasize that we never claimed to address general personalization in this paper, despite our framework being inherently adaptable to various objects by design**. As highlighted in our abstract, our work specifically targets “face blindness.” We chose to focus on humans because personalization in this context has the most widespread application. For other objects requiring personalization, users can adapt our proposed framework to their specific use cases.
> - ***Comparison with Closed-Source Models***: **We argue the fact that our proposed strategy enables a 7B open-source MLLM， which does not possess personalization capability before trained with our method, to achieve performance comparable to GPT-4o is indeed a significant contribution**. GPT-4o is a closed-source model with insufficient information available about its underlying technology. If models like OpenAI’s o1 and GPT-4o are always used as baselines, it would render many research contributions less impactful. We wish to note that our method is generic and applicable to any MLLMs and enable them with personalized capability, and would greatly appreciate it if you could consider the impacts and insights our methods can bring to small scale and open-source community.
> - ***Key Contributions***: We wish to emphasize that our primary contribution lies in proposing the in-context learning paradigm to address the MLLM personalization task and designing a data synthesis framework to create personalized training data. We do not claim that the personalized wrapper tokens are our main contribution, they are an important design element to separate each personalized concept. We also argue that the research community today prioritizes the development of generalist models rather than task-specific ones. Consequently, **taking a data-centric perspective, minimizing modifications to model architecture and training paradigms is advantageous, as it allows training to remain unified across different tasks**.
>
> We hope that we addressed your newly raised concerns, if you have any other questions, we are happy to discuss further.

---

> > ### Comment · Reviewer_YLzK · 2024-11-29
> >
> > Even though this paper enable 7b model with some personalization ability, still, it is not what claimed in the paper as SOTA performance. And those close-end models are not compared in the paper. Also, the paper lacks detailed analysis as mentioned. Therefore, I will keep my score.

---

### Author Response · Authors · 2024-11-22
**General Response**

We sincerely appreciate the time and effort the reviewers have dedicated to evaluating our paper. Their insightful advice and constructive comments have significantly contributed to improving our manuscript and addressing potential areas of confusion.

**We are grateful that the reviewers recognized the novelty and practicality of our proposed data generation framework (C8KK, ZV1c, dSk8), the value of the benchmark we provide (C8KK, ZV1c, dSk8, YLzK), our experimental results are promising (C8KK, ZV1c, YLzK), and the quality of our presentation (dSk8, YLzK).**

During the rebuttal period, we carefully addressed each concern raised by the reviewers and made the necessary adjustments to our manuscript. We hope that our responses resolve the issues and provide clarity where confusion may have arisen.

---

### Author Response · Authors · 2024-11-23
**Looking forward to further discussion**

Dear Reviewers,

We sincerely appreciate the time and effort you have dedicated to evaluating our paper. Your insightful advice and constructive comments have been invaluable in improving our manuscript and addressing areas of potential confusion. As the discussion deadline approaches, we would be grateful to receive any additional feedback you may have. Your input would greatly assist us in further enhancing the manuscript and clarifying any remaining issues.

---

### Meta-Review · Area_Chair_s5u3 · 2024-12-20

**Metareview:**

This paper addresses the "face blindness" limitation in multimodal large language models (MLLMs) by introducing Personalized Visual Instruction Tuning (PVIT), a novel framework for enabling personalized dialogues about specific individuals in images. The key scientific contribution is a data-centric approach that formulates personalization as an in-context learning problem, requiring only one-time training rather than per-individual fine-tuning. The authors develop an automated pipeline for generating personalized training data by leveraging visual experts, image generation models, and large language models. They also introduce P-Bench, a new benchmark for evaluating personalization capabilities. Experiments demonstrate that their approach significantly improves personalization performance of open-source MLLMs, achieving accuracy comparable to closed-source models like GPT-4V on their benchmark.

**Additional Comments On Reviewer Discussion:**

The paper's main strengths lie in its practical and generalizable approach to personalization through data curation rather than architectural changes, making it more accessible for real-world applications. The automated data generation pipeline and evaluation benchmark are valuable contributions to the field. However, some notable weaknesses include: limited scope focusing primarily on human faces rather than general object personalization, potential concerns about data hallucination and quality control in the automated pipeline, and incomplete analysis of failure cases particularly for challenging scenarios like similar-looking individuals or significant age differences. The paper has received generally positive reviews with scores ranging from 5-8, with reviewers particularly appreciating the practical utility and novel formulation of the problem, though some concerns remain about the scope of evaluation and ethical considerations around using human images. As for the evaluation and comparison with closed-source models/systems like GPT-4o, the AC agrees it is unfair for open-source research to always compare with them as baselines. We should advocate open science instead. But for failure case analysis, the authors should include and discuss more, saying it is challenging to do so seems not a good excuse unless the model is almost perfect (which should not be the case).

---

### Decision · Program_Chairs · 2025-01-22

Accept (Poster)